# Characterizing the social media footprint of general surgery residency programs

Erin M. White[1][☯], Stefanie C. Rohde[2][☯], Nensi M. Ruzgar[2][‡], Shin Mei Chan[2][‡], Andrew C. Esposito[1][‡], Kristin D. Oliveira[1], Peter S. Yoo[1]*

1 Department of Surgery, Yale University, New Haven, Connecticut, United States of America, 2 School of Medicine, Yale University, New Haven, Connecticut, United States of America

☯ These authors contributed equally to this work.
‡ These authors also contributed equally to this work.
* peter.yoo@yale.edu

## Abstract

### Background

The medical community has increasingly embraced social media for a variety of purposes, including trainee education, research dissemination, professional networking, and recruitment of trainees and faculty. Platform choice and usage patterns appear to vary by specialty and purpose, but few studies comprehensively assess programs' social media presence. Prior studies assessed general surgery departments' Twitter use but omitted additional social media platforms and residency-specific accounts.

### Objective

This study sought to broadly characterize the social media footprint of U.S. general surgery residency programs.

### Methods

Using a protocolized search of program websites, social media platforms (Twitter, Facebook, Instagram, LinkedIn), and internet search, cross-sectional data on social media usage in March 2020 were collected for programs, their affiliated departments, their program directors (PDs), and their assistant/associate PDs (APDs).

### Results

318 general surgery residency programs, 313 PDs, and 296 APDs were identified. 47.2% of programs had surgery-specific accounts on ≥1 platform. 40.2% of PDs and APDs had ≥1 account on Twitter and/or LinkedIn. Program type was associated with social media adoption and Twitter utilization, with lower usage among university-affiliated and independent programs (p<0.01).

### Conclusions

Most general surgery residencies, especially non-university-based programs, lacked any department or residency accounts across Twitter, Facebook, and Instagram by March

**Data Availability Statement:** All relevant data are within the manuscript and its Supporting Information files.

**Funding:** The author(s) received no specific funding for this work.

**Competing interests:** The authors have declared that no competing interests exist.

2020. These findings highlight opportunities for increased social media engagement and act as a pre-pandemic baseline for future investigations of how the shift to virtual trainee education, recruitment, conferences, and clinical care affect social media use.

## Introduction

The medical community has increasingly embraced social media for a wide variety of purposes, including dissemination of research [1] patient education [2], professional networking [3,4], and brand development [2,5]. Departments use social media accounts for promoting research, increasing visibility within the academic community, and recruiting residents and faculty [6]. For residency programs, social media plays a role in trainee education [7], and in recruiting prospective applicants [8]. Residency applicants use social media to evaluate programs [8–12], often affecting their decision to apply to a program (12%-24%), to accept an interview invitation (25%), or how to rank a program (20–29%) [8–10].

Rates of social media adoption vary by platform and specialty. Twitter has been widely adopted by Emergency medicine residencies (65% by February 2016) [13] and urology departments (49% by May 2019) [14]. Instagram enjoys widespread adoption by plastic surgery residencies (57% by June 2019) [15]. Facebook was the most prevalent platform in use by dermatology [16] and otolaryngology [17] residencies in 2017–2018, although only a quarter had accounts. Departments of Surgery that have an affiliated residency have increasingly joined Twitter, growing from 12% in January 2017 [18] to 25% by February 2019 [6]. Their use of other social media platforms has not yet been described. Likewise, while the presence of residency-specific accounts has been studied within other specialties, it has not been described in general surgery.

Because a residency program's social media presence extends beyond its departmental account–to residency program-specific accounts, program leadership's accounts, and additional social media platforms–this study sought to employ a more inclusive strategy for assessing programs' use of social media. We report here characteristics of the social media footprint of residency programs within one specialty, general surgery.

## Methods

A list of U.S. general surgery residency programs was generated from the Association of American Medical College's (AAMC) Electronic Residency Application Service [19].

Cross-sectional data were manually compiled using a protocolized search strategy (Fig 1) in March 2020. Residency program websites were first queried for program type (university, university-affiliated, independent), PD and APD names, and social media links. Twitter, Instagram, Facebook, and LinkedIn were then searched for accounts belonging to each of the following:

1. Department of Surgery

2. Residency program

3. Affiliated hospital or medical school

4. PDs and APDs.

For each account, data collected included account name, type of account, and days since most recent post. To ensure no accounts were missed, ≥2 research team members conducted

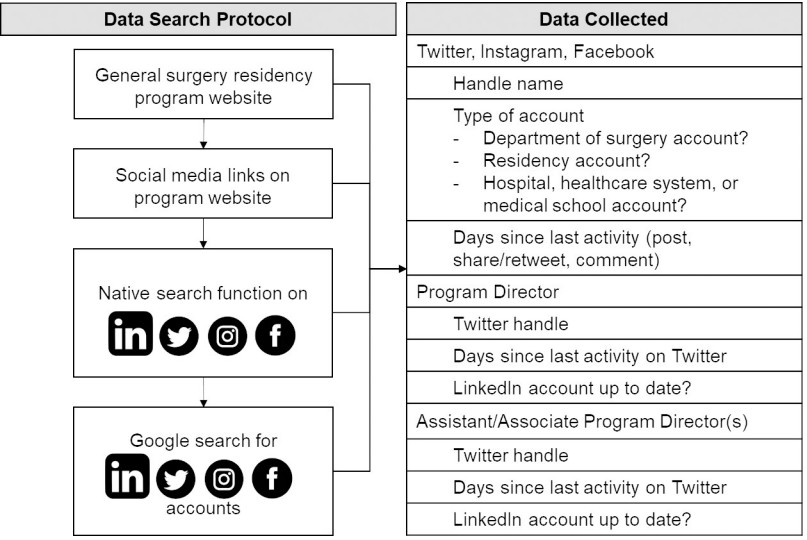

**Fig 1. Search strategy protocol.** A detailed stepwise protocol was used to guide data collection. Searches were first conducted using the complete name of a residency program; abbreviated versions of program names were also searched as needed. Abbreviated names were found on the program websites (e.g., "UAB" for University of Alabama at Birmingham, "MGH" for Massachusetts General Hospital). Program Director and Assistant or Associate Program Director accounts were found using complete name only; identity was confirmed using descriptions in their profile as a surgeon and/or their place of work.

a search for each program. Data were analyzed with summary statistics, chi-square test, and Wilcoxon rank sum test.

This study was granted exemption by the Yale University Institutional Review Board (Protocol ID:2000029886). These data were collected using publicly available services only and were accessed in accordance with the terms, conditions, and rules for each social media platform.

## Results

318 general surgery residency programs, 313 PDs, and 296 APDs were identified.

### Programs' social media use

83.6% (266/318) of program websites displayed ≥1 social media hyperlink, however the majority linked to an affiliated hospital or medical school account. Only 11.0% (35/318) linked to a surgery-specific (i.e., department or residency) social media account. The study's search protocol, however, revealed that 47.2% (150/318) do have ≥1 surgery-specific account (Table 1).

Twitter was the most commonly-used platform, with 38.7% (123/318) of programs having ≥1 surgery-specific account. 18.9% (60/318) had only a departmental account, 12.3% (39/318) had only a residency account, 6.0% (19/318) maintained separate departmental and residency accounts, and 1.6% (5/318) had a single combined account (S1 Fig). On Instagram, 15.7% (50/318) had ≥1 surgery-specific account while on Facebook, only 23.0% (73/318) did.

52.8% (168/318) of programs had no surgery-specific accounts on any platforms (Table 1). All but one, however, had representation via an account for the affiliated hospital, healthcare system, and/or medical school.

Across all platforms, a larger share of university-based programs had surgery-specific accounts than university-affiliated and independent programs (p<0.001, Table 1), and were

**Table 1. Distribution of social media account types by residency program type.**

| Program type (n = 318) | Surgery-specific account(s) | Hospital or medical school account only | No account | p-value (χ² test) |
|---|---|---|---|---|
| **Twitter** | **38.7% (123)** | **59.7% (190)** | **1.6% (5)** | |
| Independent (n = 63) | *14.3% (9)* | *84.1% (53)* | *1.6% (1)* | |
| University-affiliated (n = 84) | *15.5% (13)* | *81% (68)* | *3.6% (3)* | |
| University-based (n = 171) | *59.1% (101)* | *40.4% (69)* | *0.6% (1)* | *<0.001* |
| **Instagram** | **15.7% (50)** | **71.1% (226)** | **13.2% (42)** | |
| Independent | *1.6% (1)* | *77.8% (49)* | *20.6% (13)* | |
| University-affiliated | *9.5% (8)* | *69% (58)* | *21.4% (18)* | |
| University-based | *24% (41)* | *69.6% (119)* | *6.4% (11)* | *<0.001* |
| **Facebook** | **23% (73)** | **74.5% (237)** | **2.5% (8)** | |
| Independent | *7.9% (5)* | *92.1% (58)* | *0% (0)* | |
| University-affiliated | *8.3% (7)* | *88.1% (74)* | *3.6% (3)* | |
| University-based | *35.7% (61)* | *61.4% (105)* | *2.9% (5)* | *<0.001* |

Across all platforms, university-based programs showed higher rates of surgery-specific account use with p<0.001 (χ² test).

more recently active on Twitter (median 4 [IQR 1–19] days versus 26 [9–365] and 14 [2–110], respectively; p = 0.005) but not Facebook or Instagram (S1 Table).

Hospital and medical school accounts had more recent activity than surgery-specific accounts (p<0.05). Across all platforms, most hospital, and medical school accounts had posted within the past day (IQR 0–3) (S1 Table).

## PDs' and APDs' social media use

Twitter and/or LinkedIn accounts were identified for 40.2% (245/609) of PDs and APDs (S2 Table). Twitter accounts were identified for 31.4% (191/609); PDs and APDs did not differ in the proportion on Twitter nor the time since most recent tweet. LinkedIn accounts were identified for 54.7% (333/609) of PDs and APDs, but only 23.2% (141/609) were updated with current position, employer, and photograph.

## Discussion

As of March 2020, 47.2% of U.S. general surgery residencies had ≥1 surgery-specific account; 38.7% had Twitter, 23.0% had Facebook, and 15.7% had Instagram. Compared to university-based programs, university-affiliated and independent programs were significantly less likely to have surgery-specific accounts across all platforms and to have tweeted less recently. This suggests a potential disadvantage for these programs in the form of missed recruitment opportunities, especially during a virtual interview season. Likewise, there was a missed opportunity to engage applicants via program websites: 76.7% of programs with a surgery-specific social media account did not link to it from their program website.

Twitter accounts were identified for only 31.4% of 609 PDs and APDs. This rate is lower than found in prior studies of general surgery PDs: in 2015, 40% of 110 PDs self-reported a personal Twitter account [20]; in 2019, among 80 PDs whose departments had a Twitter account, 50% had identifiable Twitter accounts [6]. Because data for this study were collected by medical students, the data reported here best approximate what contemporary applicants might find. Some PDs/APDs may have accounts that are private, anonymous, or otherwise not easily associated with a program; such accounts likely have limited value as a recruitment tool.

## Limitations

What factors are driving the disparity between types of programs remain unclear. One explanation may be that department of surgery accounts are predominantly managed by marketing, administrative, or information technology staff (69%) rather than surgeons or trainees [6]. It is not known who manages general surgery residency-specific accounts, however, and collection of such data was outside the scope of this project.

At the time of data collection in March 2020, the COVID-19 pandemic was beginning to change the surgical training landscape, shifting clinical care, education, conferences, and fellowship interviews online [21,22]. It was announced that the entire 2020–2021 residency selection process would be conducted virtually in May 2020 [23]. Our team's ongoing monitoring of social media is show that increasing adoption and innovative strategies for engaging applicants [24].

## Future research

Once this period of rapid adoption and innovation plateaus, the new baseline should be quantified, and the impact on trainee education, research dissemination, and trainee recruitment should be evaluated. While this study did not evaluate content on social media accounts, though methods for doing so have been described in other studies [13,15,18]. How social media content has potentially changed in light of ongoing events and how content varies between different specialties, different platforms, and/or different kinds of accounts are all potential areas of study.

## Conclusions

By March 2020, 52.8% of U.S. general surgery residencies lacked any surgery-specific account on Twitter, Facebook, or Instagram, with lower rates of adoption among non-university-based programs. Only 40.2% of PDs/APDs had identifiable Twitter and/or LinkedIn accounts. These findings reflect missed opportunities for social media engagement by departments and residencies.

## Supporting information

**S1 Fig. Account types by platform.** Most general surgery residency programs have an affiliated surgery-specific account and/or hospital, healthcare system, or medical school account. Surgery-specific accounts include Department of Surgery accounts, residency program accounts, and accounts which were identified as combined departmental and residency accounts. Some programs had both separate departmental and residency accounts.
(TIF)

**S1 Table. Recent social media activity by account type and program type.** Recent social media activity, as measured by median (interquartile range [IQR]) days since most recent post, for independent, university-affiliated, and university-based U.S. general surgery residency programs. Row p-values (Kruskal-Wallis tests) are noted.
(DOCX)

**S2 Table. Social media presence of general surgery residency program leadership.** Social media accounts and activity for general surgery residency program directors (PDs) and assistant or associate program directors (APDs). Up-to-Date LinkedIn profiles included a current position and/or employer as well as a photograph.
(DOCX)

**S1 File. General surgery program social media data file.** Social media data (program and program leadership) collected by research team members in March 2020. Program director (PD) and assistant or associate PD (APD) names and Twitter handles are redacted for privacy and to ensure compliance with LinkedIn terms of use.
(XLSX)

## Acknowledgments

Thank you to medical students Oladimeji Oluaderounmu, Sumun Khetpal, and Shawn Ahn for their assistance in data collection.

## Author Contributions

**Conceptualization:** Erin M. White, Stefanie C. Rohde, Peter S. Yoo.

**Data curation:** Erin M. White, Stefanie C. Rohde, Nensi M. Ruzgar, Shin Mei Chan.

**Formal analysis:** Erin M. White, Stefanie C. Rohde, Nensi M. Ruzgar.

**Investigation:** Erin M. White, Stefanie C. Rohde, Andrew C. Esposito.

**Methodology:** Erin M. White.

**Project administration:** Erin M. White, Peter S. Yoo.

**Supervision:** Andrew C. Esposito, Kristin D. Oliveira, Peter S. Yoo.

**Visualization:** Shin Mei Chan.

**Writing – original draft:** Erin M. White, Stefanie C. Rohde.

**Writing – review & editing:** Erin M. White, Stefanie C. Rohde, Nensi M. Ruzgar, Shin Mei Chan, Andrew C. Esposito, Kristin D. Oliveira, Peter S. Yoo.

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
