## [Decision Letter · Decision Letter 0]

19 Apr 2021

PONE-D-21-08613

Characterizing the Social Media Footprint of General Surgery Residency Programs

PLOS ONE

Dear Dr. Yoo,

Thank you for submitting your manuscript to PLOS ONE. After careful consideration, we feel that it has merit but does not fully meet PLOS ONE’s publication criteria as it currently stands. Therefore, we invite you to submit a revised version of the manuscript that addresses the points raised during the review process.

Please address reviewer comments as much as possible. This is an interesting and important paper.

We look forward to receiving your revised manuscript.

Kind regards,

Leonidas G Koniaris, MD

Academic Editor

PLOS ONE

Journal Requirements:

Please provide a supplementary data file of tabulated information you collected for each general surgery institution/practice."

In your Methods section, please include additional information about your dataset and ensure that you have included a statement specifying whether the collection method complied with the terms and conditions for the website.

We note that you have indicated that data from this study are available upon request. PLOS only allows data to be available upon request if there are legal or ethical restrictions on sharing data publicly. For information on unacceptable data access restrictions, please see http://journals.plos.org/plosone/s/data-availability#loc-unacceptable-data-access-restrictions.

4a) If there are ethical or legal restrictions on sharing a de-identified data set, please explain them in detail (e.g., data contain potentially identifying or sensitive patient information) and who has imposed them (e.g., an ethics committee). Please also provide contact information for a data access committee, ethics committee, or other institutional body to which data requests may be sent.

4b) If there are no restrictions, please upload the minimal anonymized data set necessary to replicate your study findings as either Supporting Information files or to a stable, public repository and provide us with the relevant URLs, DOIs, or accession numbers. Please see http://www.bmj.com/content/340/bmj.c181.long for guidelines on how to de-identify and prepare clinical data for publication. For a list of acceptable repositories, please see http://journals.plos.org/plosone/s/data-availability#loc-recommended-repositories.

Please include captions for your Supporting Information files at the end of your manuscript, and update any in-text citations to match accordingly. Please see our Supporting Information guidelines for more information: http://journals.plos.org/plosone/s/supporting-information.

Reviewers' comments:

Reviewer's Responses to Questions

**Comments to the Author**

1. Is the manuscript technically sound, and do the data support the conclusions?

Reviewer #1: Yes

2. Has the statistical analysis been performed appropriately and rigorously? 

Reviewer #1: Yes

3. Have the authors made all data underlying the findings in their manuscript fully available?

Reviewer #1: Yes

4. Is the manuscript presented in an intelligible fashion and written in standard English?

Reviewer #1: Yes

5. Review Comments to the Author

Reviewer #1: In this study authors performed a national assessment of social media platforms to garner presence for departments of surgery and their residencies. They found that about only half of surgery residencies had some sort of social media presence, laying the way for potential improvements in social media presence for surgery residencies and departments of surgery alike. This reviewer has the following questions

1-Social media platforms are constantly evolving. What was once popular several years ago may not be popular now. How do departments and residencies effectively invest in creating and maintaining these platforms when they may be obsolete in a few years.

2-Tik Tok and SnapChat are also very popular social media platforms that the younger generations have readily adopted. Many universities are adopting these platforms as well. A quick internet search can lead to a list of Universities with TikTok pages. It would be good to include these platforms in your analysis as well.

3-In addition to program directors and assistant program directors, department chairs are also influential in their presence. This reviewer would include those individuals in analysis as well

4-Can the authors provide any type of objective data on how well the message is being received. If residencies have these accounts, is there a way to assess how often they are being seen. Adding this type of data would make the paper much stronger

6. PLOS authors have the option to publish the peer review history of their article (what does this mean?). If published, this will include your full peer review and any attached files.

Reviewer #1: No

---

## [Author Response · Author response to Decision Letter 0]

29 May 2021

Dear PLOS ONE Reviewers and Editors,

Thank you for taking the time to review our paper and providing these insightful comments. Please see below our point-by-point response to the editorial and reviewer statements. 

“2. Please provide a supplementary data file of tabulated information you collected for each general surgery institution/practice."

We have done this.

“3. In your Methods section, please include additional information about your dataset and ensure that you have included a statement specifying whether the collection method complied with the terms and conditions for the website.”

We have added a sentence to address this. See line 78-79 of the manuscript.

“1-Social media platforms are constantly evolving. What was once popular several years ago may not be popular now. How do departments and residencies effectively invest in creating and maintaining these platforms when they may be obsolete in a few years.”

We thank the reviewer for raising this issue. Answering this question is very important to residency programs because it speaks to some of the barriers to cultivating an effective social media presence. Unfortunately, the data collected in this study do not specifically address this facet of social media usage, so we elected not to provide any conjecture surrounding this question in our discussion. We believe that an appropriate strategy will be developed over time as departments and residencies gain experience with social media.

“2-Tik Tok and SnapChat are also very popular social media platforms that the younger generations have readily adopted. Many universities are adopting these platforms as well. A quick internet search can lead to a list of Universities with TikTok pages. It would be good to include these platforms in your analysis as well.”

While we agree these platforms may be up and coming platforms for future recruitment seasons, we did not include these in this study for a few reasons. Perhaps most tellingly, no programs listed Tik Tok or SnapChat on their websites as a way to connect via social media. This suggests that few, if any, programs are using these platforms as part of their marketing or recruitment strategy yet. However, the reasoning behind our decision extended beyond this. Content on the SnapChat platform is typically only visible temporarily; it also requires account holders to “friend” an account in order to access any posted content – so it is not truly public, and content posted prior to becoming a “friend” cannot be viewed retroactively. Therefore, it was not feasible to apply our study protocol, even with significant modifications, to SnapChat. Furthermore, the contribution of these platforms to programs’ digital footprints was felt by our group to be quite minimal at this point. On SnapChat, for example, searching “school of medicine” accounts yields only 7 results even as of today, and “residency” yields no apparent GME training accounts. Similarly, Tik Tok was not included in the study protocol because at the time of data collection (March 2020), it had not been widely adopted in the United States and was predominantly used by individuals under 20 years old. By some reports, the number of users more than tripled during the early part of 2020 with a shift towards an older demographic. When we repeat this study, we agree it would certainly be appropriate to include Tik Tok. This could not have been predicted at the time of our data collection and unfortunately, there is no way to retroactively determine which accounts already existed at that time; the date of account creation is not publicly available. 

“3-In addition to program directors and assistant program directors, department chairs are also influential in their presence. This reviewer would include those individuals in analysis as well”

We agree with the reviewer that department chairs who are active on social media may also contribute to a program’s social media presence. However, we chose not include them for two primary reasons. First, many residency programs rotate their trainees through multiple hospitals, which often have different department chairs. Residents from our own program, as an example, interact with three chairs of surgery, however only one is primarily involved with the recruitment and interview process. We would not have had the insider perspective at other programs to determine which chairs should or should not be included as a representative of an affiliated residency program. Second, to feasibly conduct this study, we needed to define the scope of our study protocol. The argument to include department chairs can easily be extended further to include residents currently in each program, division chairs, and even faculty who could be considered “influencers” on social media. While the full social media footprint of a given program realistically includes all these people and more, a clear and objective limit had to be defined to conduct the study. For these reasons, we defined our study protocol to only include those individuals who are officially identified as leaders within the residency program. 

“4-Can the authors provide any type of objective data on how well the message is being received. If residencies have these accounts, is there a way to assess how often they are being seen. Adding this type of data would make the paper much stronger”

Thank you for this feedback. Our group is currently exploring additional methods of mining social media data for metrics that might provide insight into this question. Metrics such as “likes” or “follows” could potentially be used as an indication of the reach a social media account has. These are imperfect metrics, however, in large part because there would be no way to parse out whether “likes” or “follows” are being generated by interactions with the target audience. In fact, a medical student in our research group recently published a perspective describing how hesitant applicants are to interact in any visible way with residency social media accounts out of concern for being judged by residency programs. Moreover, it is not possible to retroactively collect such data representing the pre-pandemic baseline. The number of followers can be collected today, but we cannot determine the number of followers as of March 2020. Similarly, the number of likes can be captured as of today, but it would not be representative of the data collected in March 2020. More than likely, it would be skewed due to the increased social media presence and reach of departments and residency programs (gained during the pandemic year’s virtual application season and activities). Given the questionable validity of these metrics to the research question of pre-pandemic social media use, we did not pursue them as part of our study design. Our future work will certainly delve further into this question.

Thank you again for the opportunity to resubmit our work. We look forward to your decision. Please feel free to contact us with additional requests or comments.

---

## [Editor Report · Decision Letter 1]

14 Jun 2021

Characterizing the Social Media Footprint of General Surgery Residency Programs

PONE-D-21-08613R1

Dear Dr. Yoo,

We’re pleased to inform you that your manuscript has been judged scientifically suitable for publication and will be formally accepted for publication once it meets all outstanding technical requirements.

Kind regards,

Leonidas G Koniaris, MD

Academic Editor

PLOS ONE
---

## [Editor Report · Acceptance letter]

22 Jun 2021

PONE-D-21-08613R1 

Characterizing the social media footprint of general surgery residency programs 

Dear Dr. Yoo:

I'm pleased to inform you that your manuscript has been deemed suitable for publication in PLOS ONE. Congratulations! Your manuscript is now with our production department. 

Kind regards, 

on behalf of

Dr. Leonidas G Koniaris 

Academic Editor

PLOS ONE